# In Vitro Study of the Effect of Ensiling Length and Processing on the Nutritive Value of Maize Silages

**DOI:** 10.3390/ani13030344

**Published:** 2023-01-19

**Authors:** Ana Gordo, Belén Hernando, Jesús Artajona, Manuel Fondevila

**Affiliations:** Departamento de Producción Animal y Ciencia de los Alimentos, Instituto Agroalimentario de Aragón (IA2), Universidad de Zaragoza-CITA, M. Servet 177, 50013 Zaragoza, Spain

**Keywords:** time of ensiling, dehydration, pelleting, in vitro fermentation

## Abstract

**Simple Summary:**

Ensiling whole plant maize is a widely used procedure to preserve forage for feeding dairy cows. The ensiling process is extended for 2 to 12 months, but it has been suggested that a long ensiling length increases the starch availability by protein solubilisation. Other ways oof processing maize silages by dehydration or pelleting may stabilise the nutritive value and facilitate transportation, despite its increased costs. Therefore, three ensiling lengths (3, 6, and 9 months) and three processing forms (fresh, dried, and pelleted) of maize silage were compared in terms of their effects on in vitro rumen fermentation and digestibility. No major differences of ensiling lengths were detected in the chemical composition of silages, but microbial fermentation was reduced when extended to 9 months. Both dehydration and pelleting increased the pH of maize silage in 0.7 units compared to fresh silage. However, the response in fermentation to processing maize silages depends on the ensiling length and only manifested in the 3 month samples. Both dehydration and pelleting tended to reduce the methane concentration in the gas produced from fermentation, indicating both a more efficient fermentation and a lower potential of greenhouse gas emissions than the use of fresh silage.

**Abstract:**

The effect of the ensiling length (3, 6, or 9 months), and the processing by dehydration (D) or dehydration and pelleting (P) with respect to the fresh silages (F) were studied in vitro on three maize cultivars in three incubation runs to study the effect of these factors on the nutritive value of maize silage. Gas production pattern, in vitro true digestibility (IVTD), methane concentration (6 and 12 h), ammonia, and volatile fatty acid concentration (VFA) at 12 h were measured. The moisture and pH of F averaged 676 g/kg and 4.09, respectively, and were not affected by the ensiling length, but moisture was reduced, and the pH increased in D and P with respect to F (*p* < 0.05). The ensiling length did not affect the chemical composition, but differences among the processing forms were detected in a higher acid detergent insoluble nitrogen (ADIN) proportion in P than D, and D than F (*p* < 0.001). Silages opened at 9 months showed the lowest gas production (*p* < 0.05), and those that opened after 3 months showed the highest IVTD. The effects of processing on nutrient utilisation only manifested on 3 month silages, with the volume of gas production and IVTD being lower in D than F. However, processing tended (*p* = 0.064) to reduce the methane proportion at 12 h, indicating both a more efficient fermentation and a lower potential of greenhouse gas emissions compared to the fresh silages. Extending the length of ensiling to 9 months reduced the fermentation of maize silage. The processing increased the dry matter and buffered the feed as well as contributed to an increase in fermentation in 3 month silages.

## 1. Introduction

The storage of harvested biomass is a key issue for ruminant production since feed availability is limited by the amount of available pasture. Ensiling is often chosen [1,2] as it is an effective procedure for preserving feed with minimum nutrient losses. Because of its high productivity, energy value, and easy processing, whole plant maize (*Zea mays*) is commonly chosen for ensiling, especially for milk production livestock [3,4]. Several agronomic factors and ensiling conditions that may affect forage quality have been studied [5,6,7], among them the ensiling length. Usually, maize silages are maintained anaerobically for 2 to 12 months before being fed, as some microbial processes still occur during extended silage times. For example, one effect of starch availability for rumen microbiota is related to the long-term solubilisation of the protein, where starch is embedded by the action of bacterial activity and maize enzymes [8,9]. Thus, whereas the rate and extent of the fermentation of non-soluble components of maize were not affected in the first 120–180 days of ensiling [10,11], ensiling for 240–365 days may decrease fibre and increase proteolysis [8,12]. However, the literature shows variable results in this regard, and Aloba et al. [13] did not observe any effect on the digestibility or metabolisable energy content of ensiling sorghum for 180 day.

The quality of silages may be reduced after opening because of the activity of aerobic organisms that increases the pH and temperature as well as textural changes that may affect the palatability and intake [14]. Furthermore, the growth of certain fungi may promote the production of mycotoxins that challenge animal health [15]. Additionally, the high moisture proportion of silages (up to 700 g/kg) makes its transportation and commercialisation difficult. The processing of maize silage by dehydration may be a way to stabilise the nutritive value of the forage and facilitate its transport and distribution. Rapid drying at high temperatures in a rotary drum stimulates rapid water evaporation, thus avoiding protein denaturalisation and maintaining its nutritive value [16], which may compensate for the combustible costs. In fact, these authors stated that dehydration reduces protein solubility and rumen ammonia concentration. Protein degradability is also reduced, increasing nitrogen input to the duodenum and protein efficiency [17]. However, Fadel [18] argued that the reduction in protein solubility is partly related to an increase in indigestible fibre-bound nitrogen. In other words, the simultaneous application of heat and pressure during pelleting disrupts the starch/protein matrix and promotes further starch gelatinisation, which might increase rumen fermentation [19,20]. In contrast, the hardness of pellets affects the microbial accessibility and reduces its utilisation [21]. However, the availability of published scientific information on processing maize silage by dehydration and pelleting is very scarce.

In this work, the effects of the length of the ensiling of maize silage and its processing (dehydration or dehydration and pelleting) for increasing stability, together with the potential interaction of these two factors, were studied in terms of the chemical composition and in vitro rumen microbial fermentation. The in vitro gas production technique has been previously applied for the assessment of the nutritive value of silages [6,10,11].

## 2. Materials and Methods

### 2.1. Substrates

Samples of three maize cultivars, two genetically modified crops (Pioneer N016-124P and N016-134P) and one isogenic (Pioneer N016-073), harvested in autumn (October/November 2020) from the location of Buñuel (Navarra, Spain; 41°58′50″ N, 1°26′36″ W), were used in the study (Table 1) and considered as replicates for the study of the main effects. The three maize cultivars did not differ in chemical composition. Maize plants at their grain milk stage (300 to 350 g dry matter, DM, per kg) were chopped to 1.5 cm (JD 9800 chopper, John Deere) and ensiled as a stack on the ground (350 t) with a lactic acid bacteria inoculant (Bon Silage Fit M, Bon Silage, Germany, 200 g per silage), with temperatures ranging from 15 to 25 °C. Silages were opened at three different ensiling lengths: 92 (3 months), 182 (6 months) and 270 (9 months) days. Once opened, each silage was divided into three equal parts. The first part was used in its original state (fresh, F) and the remaining two-thirds of each silage were dehydrated for around 15 min at 650–750 °C in an LN400 industrial drying system, until reaching a dry forage with 12% moisture (D). Then, half of the dried silage was ground to 4 mm and pelleted to a 8 mm pellet size (P). Five samples (0.5 kg) from each silage and processing form were collected for each of the 27 substrates (three silages opened at three ensiling lengths, in three presentation forms, from the three maize crops) for analytical purposes. Once dried (60 °C, 48 h), the samples were ground to a 1 mm particle size in a hammer mill (Retsch Gmbh/SK1/417449, Haan, Germany) for chemical analysis and in vitro incubation, and the average of the five samples for each treatment were considered.

### 2.2. Incubation Procedures

Rumen inoculum was obtained through the rumen cannula of three adult sheep from the Animal Experimentation Service of the University of Zaragoza. Sheep were given daily 500 g of a 1:1 mixture of alfalfa hay and barley straw plus 500 g of a concentrate mixture (300 g barley grain, 100 g maize grain, and 100 g soybean meal) for 14 days before the start of the trial. Before the morning offer of feed (9:00), rumen contents (approximately 300 mL) were withdrawn from each animal and filtered through a cheesecloth, mixed, dispensed in thermos flasks, and immediately transferred to the lab for incubation. Animal care and procedures for the extraction of rumen inoculum were approved by the Ethics Committee for Animal Experimentation (protocol PI48/20). The care and management of animals agreed with the Spanish Policy for Animal Protection RD 53/2013, which complies with EU Directive 2010/63 on the protection of animals used for experimental and other scientific purposes.

A pooled sample of each substrate (n = 27) was incubated in vitro in a closed batch system, following the procedures by [22], but without microminerals and resazurin. Three consecutive incubation series were carried out for 48 h at 39 °C under anaerobic conditions, and the incubation pH was adjusted to 6.3 according to [23]. Triplicate glass bottles (116 mL total volume) were filled with 800 mg of sample sealed in a nylon bag (50 µm pore size) and then 80 mL of the incubation medium including rumen inoculum at a proportion of 0.2 of the total incubation volume (16 mL). Three additional bottles without substrate were also included as blanks. Two incubated bottles were used for measuring the gas pressure as an index of microbial fermentation for 2, 4, 6, 8, 10, 12, 24, and 48 h. Pressure was recorded with a HD 2124.02 manometer fitted with a TP804 pressure gauge (Delta Ohm, Caselle di Selvazzano, Italy). Readings were converted into volume by a pre-established linear regression equation between the pressure recorded in the same bottles under the same conditions and known air volumes (n = 103; R^2^ = 0.996), and the results were expressed per unit of incubated organic matter (OM). At the end of the incubation, the residues were washed and dried (60 °C, 48 h) for the determination of dry matter disappearance (DMd). Residues from the three series were pooled by sample and treated with neutral detergent to determine the in vitro true digestibility (IVTD) of the substrates as the proportion of the difference between DM incubated and the residual NDF, as indicated by [24].

The third incubated bottle of each sample was used to determine the methane production and concentration at 6 and 12 h from its gas phase. Furthermore, the liquid incubation medium was sampled at 12 h and stored frozen until the determination of the ammonia (2 mL sample on 2 mL hydrochloric acid) and volatile fatty acid concentration (VFA, 0.5 mL sample on 2 mL of 0.5 M orthophosphoric acid with 1 mg 4-methyl-valeric acid as internal standard).

### 2.3. Analytical Procedures

The AOAC methods [25] were used for the analysis of DM (ref. 934.01), OM (ref. 942.05), crude protein (CP, ref. 976.05), and ether extract (EE, ref. 2003.05) contents in the silage samples. Concentration of neutral detergent fibre (aNDFom) was analysed as described by [26] in an Ankom 200 Fibre Analyser (Ankom Technology, New York, NY, USA) using α-amylase and sodium sulphite, and the results were expressed exclusive of the residual ashes. The acid detergent fibre (ADF, ref. 973.18) and acid detergent lignin (ADL) were determined as described by [25,27], respectively. The nitrogen linked to the ADF residue (acid detergent insoluble nitrogen, ADIN) was also analysed. Total starch content was determined enzymatically from samples ground to 0.5 mm using a commercial kit (Total Starch Assay Kit K-TSTA 07/11; Megazyme, Bray, Ireland).

The pH of the ground silage samples was determined at the laboratory with a CRISON micropH 2001 (Barcelona, Spain) after 30 min of suspending a 2 g sample in 20 mL of distilled water. Methane concentration was measured in an Agilent 6890 apparatus equipped with an FID detector and a capillary column (HP-1, 30 m × 535 µm id × 1.5 µm film thickness), calibrated with a 10% CH_4_ standard, with a flux of 2 mL/min at 250 °C. Injection was in split mode, with He as the carrier gas. The frozen samples of incubation medium were thawed and centrifuged at 13,000× *g* for 15 min at 4 °C for their analysis of VFA, which was determined by gas chromatography on the same apparatus as for the methane analysis, with a capillary column (HP-FFAP Polyethylene glycol TPA, 30 m × 530 μm id). Injection was in a column, with He as the carrier gas and a temperature ramp from 80 to 165 to 230 °C. The concentration of ammonia was determined colorimetrically [28].

### 2.4. Statistical Analysis

Results were analysed statistically by ANOVA with the Statistix 10 package [29], according to the model: *Y* = *μ + Li + Pj + LPij + εijk*, with the ensiling length (*Li*, *i* = 3) and the processing form (*Pj*, *j* = 3) as the main factors, and these and their interactions were contrasted with the residual error (*εijk*). Because of the lack of differences among the maize cultivars, this was not included as a block in the model and was considered as the experimental unit (*k* = 3) and included in the error term.

For the in vitro fermentation studies, the incubation series was included as a block, and the average of the two bottles of each treatment in each series was considered as the experimental unit. Treatment differences among means with *p* < 0.05 and 0.05 < *p* < 0.10 were accepted as representing statistically significant differences and a trend to differences, respectively. When significant, differences were contrasted by the Tukey test.

## 3. Results

Silage moisture proportion when opened (F) was, on average, 676 ± 23.0 g/kg, whereas after processing, it was 145 ± 29.0 and 133 ± 16.3 g/kg for the D and P silages, respectively, without recording differences among the lengths of ensiling (*p* > 0.10). The pH of the maize silages ranged from 4.02 to 5.13 (average pH 4.43) and was affected by processing (*p* = 0.026), being lower in F than P (average values of 4.09, 4.40, and 4.81 for F, D and P, respectively; SEM = 0.169). However, there were no effects of ensiling length (*p* = 0.612) nor in the interaction length × processing (*p* = 0.832) on pH.

No effects of the ensiling length were recorded in the chemical composition (Table 2). In contrast, processing affected the nutrient proportions, being higher in F than P for OM (*p* = 0.004) and D for EE (*p* = 0.010) concentration, whereas the CP and starch content were unaffected. Regarding the characterisation of the fibrous fraction, higher proportions of aNDFom (*p* = 0.013) and ADF (*p* = 0.012) were recorded in D than P, whereas ADL was higher in P than F (*p* = 0.034). Similarly, ADIN was the highest in P, and higher in D than in F (*p* < 0.001).

The final pH after the three 48 h incubation runs ranged from 6.1 to 6.4 (average value 6.25). Regarding the effect of the length of ensiling on in vitro gas production, the 9 month silages rendered the lowest volume throughout the incubation, whereas from 0 to 4 h of incubation, those opened at 6 months produced more gas (*p* < 0.05) than at 3 months (Table 3). Among the processing forms, F samples produced more gas than P at 2, 4, and 24 h of incubation, and more gas than D from 8 h onwards (*p* < 0.05). However, the interaction length × processing (*p* < 0.05 from 2 to 10 h, *p* = 0.052 at 12 h) indicates that these differences only manifested for silages opened at 3 months. In any case, the magnitude of differences in gas production among the types of processing was moderate, being on average 0.12 to 0.16 times higher in F than D and P from 12 to 48 h of incubation.

Values for both DMd as an index of 48 h in vitro rumen degradability, and IVTD as an index of total tract digestibility, were highest for the 3 month silages (DMd of 0.499, 0.476 and 0.478 and IVTD of 0.649, 0.626 and 0.626 for 3 m, 6 m and 9 m, respectively; *p* < 0.01). On the other hand, DMd was reduced by processing (Table 4), being highest in F, intermediate in P, and lowest in D (average values of 0.513, 0.450, and 0.490 for F, D, and P, respectively; *p* < 0.001). However, in terms of the IVTD differences between F and P, they did not reach significance (values of 0.653, 0.603, and 0.645, respectively; *p* < 0.001). The interaction length × processing was not significant for any of these parameters. No effect was detected in the methane concentration at 6 h, but at 12 h of processing, it tended to reduce it (Table 4), with average values of 0.871, 0.818, and 0.826 for F, D, and P, respectively (*p* = 0.064). Therefore, considering this and the total gas produced in the calculation of the volume of methane expressed per unit of incubated substrate, this was lower in D and P than in F, although the response was only manifested for silages opened after 3 months (interaction length × processing, *p* = 0.072).

The comparison of the effect of processing on ammonia concentration after 12 h of incubation (Table 5) depends on the date of opening (significant interaction length × processing, *p* = 0.002). Thus, 3 months was the highest for the F silages, whereas for 6 months, the F was higher than P and for 9 months, it was higher with D than P. Ammonia concentration increased with ensiling length, being highest for 9 months, intermediate for 6 months, and lowest for 3 months (*p* < 0.001). Regarding the total VFA concentration, differences due to processing were only observed in the 3 month silages, being higher in F than P (interaction length × processing, *p* = 0.036). Molar proportions of VFA after 12 h incubation indicated a lower acetate proportion in D than P (average values of 0.585, 0.578 and 0.593 for F, D, and P, respectively; *p* = 0.03), whereas that of propionate was lowest for F, intermediate for P, and highest for D (0.222, 0.234, and 0.244, respectively; *p* < 0.001). A reduction in butyrate proportion with processing was observed in the 3 month and 6 month silages, but not in the 9 month (interaction length × processing, *p* = 0.006). Results on valerate and BCFA proportions were randomly ranked and must be taken with caution because of the low magnitude of the values observed.

## 4. Discussion

The observed moisture proportion and pH values of the fresh maize silages were within the range reviewed by [7] for what these authors considered as normal silages (300–350 g DM/kg), although the NDF and starch proportions were 1.16- and 0.87-fold of those reported by them, respectively. However, our values for these parameters were close to those cited by [30], who pointed out that the considerable variability of NDF in maize silages and the methodological variation among laboratories could result in systematic differences in NDF content. In their review, Khan et al. [7] indicated an average in vivo OM total tract digestibility of 602 g/kg of fresh silages, which was quite close to the IVTD determined here under in vitro conditions (average of 653 g/kg, Table 4).

### 4.1. Effect of Length of Ensiling

The available information on the effect of the length of ensiling on the characteristics of maize silage can be divided into those studies that cover the first 90 or 120 days, aiming to look for the minimum period that the silage conditions must be maintained for ensuring stability [10,11], and those that look for the maximum length of ensiling in order to check for potential changes [8,12,13]. The second approach fits our experimental design in the present study. In general, minor changes have been reported in chemical composition, nor in NDF digestibility, among the times of ensiling ranging from 90–120 to 240–270 days [8,12]. Accordingly, the extension of the ensiling length from 3 months to 9 months in this work did not promote changes in pH or the chemical composition of maize silages.

However, the extension in the ensiling length from 3 to 9 months showed lower rumen microbial fermentation throughout 48 h of incubation, which was also reflected in a lower DMd and IVTD. An increase in starch digestibility with the ensiling length has been reported [8,12] and attributed to an extended digestion of zein and the increase in protein solubility [31], which release starch from the endosperm protein matrix. In fact, Der Bedrosian et al. [8] found a high relationship between the concentration of soluble protein and starch digestibility (R^2^ = 0.78). A release of starch in the rumen by protein solubilisation during ensiling should increase rumen starch degradation, and thus reduce the amount reaching the duodenum. Consequently, the change in starch digestion site should decrease the efficiency of energy utilisation. Furthermore, it has been argued that the faster microbial fermentation activity promoted by an increased degradation of starch from maize silage might contribute to a reduction in rumen pH, enhancing the risk of acidosis [8]. However, assuming a rumen starch degradation of maize silages of 0.91 [32], a major effect on the rumen environment and the rumen outflow of starch by a higher protein degradation are difficult to expect. In the present experiment, the higher ammonia concentration during the in vitro incubation of the 9 month samples (*p* < 0.001) might indicate a faster microbial protein degradation, but no differences due to ensiling length were detected among the fresh silages (interaction length × processing, *p* = 0.002). In any case, despite starch digestion not being measured in our study, our results show a reduction in in vitro gas production and 48 h DMd when the ensiling length was extended from 3 to 6 or 9 months, which was reflected in a lower IVTD.

Therefore, extending the ensiling period for maize up to 9 months does not appear to be an advantage. In contrast, minor differences were detected on ensiling for 90 or 180 days. Der Bedrosian et al. [8] observed minimal differences in the in vitro NDF and starch digestibility among maize ensiled for 90, 180, or 270 days or a non-brown midrib hybrid maize silage of DM content within the range of those studied here, and Aloba et al. [13] did not find any effect of extending the ensiling length of sorghum whole plant from 75 to 180 days on the energy concentration or digestibility, as estimated from the in vitro gas production. However, a higher in vitro starch digestibility was observed by [12] between ensiling for 270 compared with 120 days.

### 4.2. Effect of Processing

The significant length × processing interaction recorded for many chemical or fermentative parameters is explained by the fact that, in most cases, a potential effect of silage processing could be detected in the 3 months or 6 months silages, but rarely at 9 months.

Actually, several companies commercialise maize silage dehydrated at high temperatures for ruminant feeding, but to our knowledge, there is no available literature regarding the study of such processing techniques, unless for biomass or biogas production. Early on, Owen [33] simply mentioned that the feed intake in milking cows or the growth in beef steers did not differ when using maize and sorghum silages either fresh or dehydrated.

Processing maize silage by dehydration (D) or by dehydration followed by pelleting (P) increased the substrate pH by 0.3 and 0.7 units, respectively. This suggests a potential improvement in animal feed intake, since it has been stated that the intake of silages with an initial pH below 4.0 pH is enhanced when increased to over 4.5 [34]. In terms of composition, no effects on the non-fibrous fraction of silages were detected (CP, starch), and the reduction in EE by dehydration may be partly related to both the low EE proportion and a potential loss of carotenoids by heating [35]. However, a higher proportion of the fibrous fraction (aNDFom and ADF) was observed in D with respect to the P silages, which might indicate a potential solubilisation of hemicelluloses and cellulose by the application of pressure and temperature while pelleting. The content in nitrogen bound to the less digestible fraction of fibre (ADIN), being higher in both P and D than in F silages, has also been reported with heating treatments, since temperatures over 60 °C result in non-enzymatic protein–carbohydrate reactions [36]. In this regard, a 2.5-fold increase in ADIN was reported in alfalfa dehydrated at 120 °C for 24 h [37], a similar proportion to that observed here for P (2.2-fold) but higher than the increase recorded by D (1.6-fold).

Rumen fermentation of silages as indicated by the volume of gas produced in vitro showed different responses to their processing depending on the length of ensiling. Thus, for the 3 month silages, the processing either by dehydration or by pelleting after dehydration reduced the extent of gas production throughout the entire fermentation period, whereas no processing effect was observed in the 9 month samples, and only at 48 h of incubation for 6 m. Despite the length of ensiling, the processed silages also showed a lower DMd than their fresh presentation form, although differences between F and P were not significant. However, differences between F and P were detected in the 12 h VFA concentration in the 3 month samples.

After 12 h of incubation, the methane concentration in gas released from D silages tended to be lower than F. This agrees with the VFA pattern at the same incubation time, with a higher proportion of propionate at the expense of the acetate and butyrate proportions, in dehydrated with respect to fresh silages, thus showing a more efficient utilisation of fermentable energy. This response is not apparent when the total volume of methane produced is considered, as it depends on both the methane concentration in the fermentation gas and the total volume of gas produced, which for 3 months was higher in F. In any case, the lower volume of methane produced in the 3 month silages after processing per unit of incubated substrate should imply a lower emission of greenhouse gases by the livestock consuming this feed.

Regarding pelleting maize silages, the magnitude of the potential increase in starch availability for rumen fermentation [19] should be relative, considering the low proportion of starch in maize silages (on average, 295 g starch/kg, Table 2) compared to whole cereal grains or concentrates. In our experiment, the differences between pelleting and dehydration were not great. In other words, the potential reduction in the microbial fermentation rate promoted by the physical characteristics of pellets such as hardness and particle size with respect to the dehydration of silages with larger particle size, in vitro [21] and in vivo [20], could not be approached in this experiment, as all samples were incubated at the same size (1 mm). Furthermore, this effect is associated with high concentrate, low fibre diets [38], so its relevance should be minor for maize silages in all-forage diets. In terms of the fermentation characteristics, pelleted silages tended to reduce the ammonia concentration and molar proportion of butyrate at the expense of propionate in the incubation medium, in agreement with the in vivo results using concentrate feeds [20].

## 5. Conclusions

A prolonged length of ensiling for 6 or 9 months does not promote major composition differences with respect to ensiling for 3 months. Rumen fermentation measured as gas production gave similar results at 3 and 6 months of ensiling, but was lower when the ensiling length was extended to 9 months. Similarly, the highest total in vitro digestibility was observed for the 3 month silages, which reduces the interest of longer ensiling times in terms of energy utilisation. Dehydrating, with or without subsequent pelleting, increased the dry matter and consequently facilitates silage handling. These procedures did not greatly affect the chemical composition but buffered the substrate, especially pelleting, which was on average 0.7 pH units higher than fresh silage. However, the response in gas production to processing maize silages depends on the ensiling length and only manifested in the 3 month samples. Dehydration and pelleting reduced, to a certain extent, the microbial fermentation of the 3 month silages since gas production was lowered throughout the incubation period. However, processing tended to reduce the proportion of methane in total gas, indicating both a more efficient fermentation and a lower potential of greenhouse gas emissions than the use of fresh silages.

## Figures and Tables

**Table 1 animals-13-00344-t001:** Description of the maize crop samples and dates of ensiling (2020) and opening (2021) of silages.

Cultivar	Date of Ensiling	Opening Date 3 m	Moisture g/kg	Opening Date 6 m	Moisture g/kg	Opening Date 9 m	Moisture g/kg
N016-134P	23/11	24/02	704	18/06	692	26/08	713
N016-124P	27/10	26/01	683	26/04	669	26/08	643
N016-073	29/10	27/01	663	26/04	661	26/08	660

**Table 2 animals-13-00344-t002:** Chemical composition (g/kg DM) of silages opened after 3, 6, and 9 months, before (F) and after heat processing (dehydrated, D, or dehydrated and pelleted, P).

Ensiling Length	Process	OM	CP	EE	NDF	ADF	ADL	ADIN	Starch
3 m	F	960	66	23	432	229	10.4	11.7 c	301
6 m	F	960	73	22	450	236	8.7	12.7 c	295
9 m	F	960	73	20	459	250	10.3	12.0 c	290
3 m	D	953	68	14	446	246	14.5	21.2 b	325
6 m	D	954	73	15	484	270	10.8	18.4 bc	286
9 m	D	951	76	18	509	282	11.7	18.4 bc	202
3 m	P	949	70	16	382	215	14.8	24.6 ab	391
6 m	P	943	85	19	434	240	15.7	31.1 a	292
9 m	P	946	74	22	403	222	11.1	24.8 ab	314
SEM		4.6	5.9	2.2	26.7	15.0	1.74	1.36	51.8
*P-*length		0.875	0.180	0.333	0.186	0.187	0.318	0.152	0.260
*P*-processing		0.004	0.536	0.010	0.013	0.012	0.034	<0.001	0.363
*P-*interaction		0.945	0.786	0.432	0.824	0.765	0.371	0.036	0.720

OM, organic matter; CP, crude protein; EE, ether extract; NDF, neutral detergent fibre; ADF, acid detergent fibre; ADL, acid detergent lignin; ADIN, acid-detergent insoluble nitrogen. SEM: standard error of means for the length × processing interaction (n = 9). Within columns, letters indicate significant differences among means (*p* < 0.05).

**Table 3 animals-13-00344-t003:** In vitro gas production (mL/g OM) as an index of the fermentation pattern of silages opened after 3, 6, and 9 months, before (F) and after heat processing (dehydrated, D, or dehydrated and pelleted, P).

Ensiling Length	Process	2 h	4 h	6 h	8 h	10 h	12 h	24 h	48 h
3 m	F	8.8 a	20.2 a	33.6 a	47.9 a	62.8 a	78.0	112.3	153.7
6 m	F	8.1 ab	18.4 ab	30.8 ab	44.2 ab	58.5 abc	72.5	104.2	145.3
9 m	F	4.5 d	12.9 d	23.6 d	35.9 c	49.1 d	63.0	94.4	132.9
3 m	D	6.3 bcd	15.4 bcd	26.7 bcd	39.2 bc	53.0 cd	66.5	94.1	127.1
6 m	D	7.5 abc	17.1 abc	28.8 abc	41.3 abc	54.0 bcd	66.7	91.8	122.7
9 m	D	5.7 cd	14.3 cd	25.1 cd	35.6 c	46.0 d	57.2	82.2	114.9
3 m	P	4.8 d	13.6 cd	24.7 cd	38.2 bc	52.8 cd	67.3	96.2	131.0
6 m	P	8.1 ab	18.9 ab	31.4 ab	46.0 a	61.1 ab	75.7	101.5	134.2
9 m	P	5.5 cd	13.6 cd	23.6 d	35.3 c	48.1 d	61.3	91.6	126.2
SEM		0.46	0.82	1.14	1.47	1.80	2.28	2.94	3.24
*P-*length		<0.001	<0.001	<0.001	<0.001	<0.001	<0.001	<0.001	<0.001
*P-*processing		0.030	0.017	0.007	0.005	<0.001	<0.001	<0.001	<0.001
*P-*interaction		<0.001	<0.001	<0.001	0.002	0.011	0.052	0.157	0.168

SEM: standard error of means for the length × processing interaction (n = 9). Within columns, letters indicate significant differences among means (*p* < 0.05).

**Table 4 animals-13-00344-t004:** Dry matter disappearance after 48 h (DMd), in vitro true digestibility (IVTD), and methane concentration (µmol/mL gas) and production (µmol/g OM) at 6 and 12 h of incubation of silages opened after 3, 6, and 9 months, before (F) and after processing (dehydrated, D, or dehydrated and pelleted, P).

Ensiling Length	Process	DMd	IVTD	Methane Concentration		Methane Production	
				6 h	12 h	6 h	12 h
3 m	F	0.533	0.668	0.456	0.889	14.7 a	68.6
6 m	F	0.503	0.645	0.439	0.865	13.5 ab	63.2
9 m	F	0.503	0.646	0.411	0.859	9.6 c	54.0
3 m	D	0.464	0.620	0.408	0.820	10.6 bc	54.2
6 m	D	0.444	0.592	0.394	0.809	13.2 ab	53.8
9 m	D	0.440	0.598	0.416	0.825	10.5 bc	47.2
3 m	P	0.499	0.660	0.447	0.824	10.8 bc	54.9
6 m	P	0.479	0.642	0.432	0.842	13.2 ab	63.0
9 m	P	0.491	0.635	0.410	0.811	9.6 c	50.0
SEM		0.0092	0.0086	0.0233	0.0293	0.73	2.48
*P-*length		0.005	0.002	0.425	0.867	<0.001	<0.001
*P*-processing		<0.001	<0.001	0.267	0.064	0.009	<0.001
*P*-interaction		0.750	0.967	0.750	0.909	0.005	0.072

SEM: standard error of means for the length × processing interaction (n = 9). Within columns, letters indicate significant differences among means (*p* < 0.05).

**Table 5 animals-13-00344-t005:** Concentration of ammonia (NH_3_, mM) and volatile fatty acids (VFA, mM), and molar VFA proportions (mol/mol) after 12 h in vitro fermentation of silages opened after 3, 6, and 9 months, before (F) and after heat processing (dehydrated, D, or dehydrated and pelleted, P).

Ensiling Length	Process	NH_3_	VFA	Acetate	Propionate	Butyrate	Valerate	BCFA
3 m	F	7.27 b	38.7 a	0.594	0.213	0.160 ab	0.010 bcd	0.023
6 m	F	7.55 ab	37.6 ab	0.574	0.228	0.163 a	0.012 ab	0.023
9 m	F	7.50 ab	36.8 ab	0.586	0.224	0.155 ab	0.011 abc	0.024
3 m	D	6.42 cd	36.7 ab	0.583	0.244	0.141 cd	0.009 cd	0.022
6 m	D	7.03 bc	37.6 ab	0.577	0.248	0.141 cd	0.010 abc	0.023
9 m	D	8.02 a	36.1 ab	0.572	0.238	0.152 abc	0.012 a	0.0226
3 m	P	6.01 d	35.0 b	0.601	0.235	0.135 d	0.008 d	0.022
6 m	P	6.52 cd	37.2 ab	0.590	0.239	0.140 cd	0.010 bcd	0.022
9 m	P	6.85 bc	37.6 ab	0.589	0.230	0.148 bcd	0.010 bcd	0.023
SEM		0.164	0.752	0.0053	0.0043	0.0030	0.0004	0.0004
*P-*length		<0.001	0.482	0.015	0.054	0.041	<0.001	<0.001
*P-*processing		<0.001	0.176	0.003	<0.001	<0.001	<0.001	<0.001
*P-*interaction		0.002	0.036	0.556	0.293	0.006	0.035	0.091

BCFA: branched-chain fatty acids proportion (sum of isobutyrate and isovalerate). SEM: standard error of means for the length × processing interaction (n = 9). Within columns, letters indicate significant differences among means (*p* < 0.05).

## Data Availability

Not applicable.

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
