# Peer review of "In Vitro Study of the Effect of Ensiling Length and Processing on the Nutritive Value of Maize Silages"

_animals, 2023, doi:10.3390/ani13030344_

Round 1

Reviewer 1 Report

This article is well written and presented. However, there are some shortcomings that should be improved before publication. 

The article is not novel but interesting scientifically, as it provides information that is not easy to find in the literature.

The most critical aspects, from my point of view are:

M&M: 3 different varieties are used to make the silages but then they are not taken into consideration, only 3 times and 3 processes are studied.

Results: The results are rather poor. All the tables are difficult to understand easily. The differences contrasted by tukey test are not clear and therefore very difficult to be interpreted.  

The dynamics of gas production could be presented, instead to Table 3. 

 The paper is well written but it is poor. The discussion is based on assumptions.

Specific aspects: 

L179-189: rewrite. Interactions should be presented first and then period and processing effects separately. 

Table 5: why is not presented the pH values? 

L314: the reduction of EE may be due to the loss of chlorophyll compounds, such as carotenoids, from corn. The pelletization and dehydartation leads to a loss of carotenoids 

Author Response

Thanks very much for your comments and suggestions.

M&M: The comparison of the three maize varieties used in the study, that were quite similar in terms of composition and degradability, was not an objective of the study. Having this into consideration, maize varieties were considered as repetitions in order to improve the strength of the in vitro design for the study of the two factors of interest: the ensiling length and the processing form of maize silages. To avoid confusions in this regard, the expression “27 treatments” (formerly in L90) has been changed to “27 substrates” in the present version.

Results: We agree with this comment regarding tables. Tables are complicated as they must be in agreement with the 3 x 3 experimental design. They show the length x processing means since this interaction resulted significant for many parameters (and differences among these means is what the Tukey test shows in tables by the different letters). Further, for giving a complete picture, the P-values for main factors and the interaction are also shown in tables.  In any case, tables presentation in this version has been modified for their improvement.

If what the reviewer suggests is to show the coefficients of the adjusted curves in Table 3, I have to disagree. Mathematical fitting of fermentation results of substrates is useful to characterise the substrates, but not to compare differences in fermentation that occur during the rumen residence time of feeds, which should be around 48 h. Further, in the case of a forage-type feed the incubation period here should not render a good mathematical adjustment to a exponential curve respect to a total extenuation of fermentation (for example, at 96 h), since an important part of microbial activity may potentially occur at later stages. This is why in our papers we prefer to show actual values during the period when rumen microbiota can develop its activity over the substrate when available (i.e., Ortolani et al., 2020, Animals 10, 732; Amanzougarene and Fondevila 2022, Animals 12, 345, among others in several other journals).

Discussion: There is a lack of information about dehydration and pelleting of maize silages (which is one of the novelties of this work), and most information on this topic is related with alfalfa, which is a totally different forage. This contributes to make the Discussion quite speculative. In any case, we have tried to improve this section.

L179-189: The fact that an interaction resulted significant indicates that the response to a specific main factor may be altered by the effect of the other. Therefore, for a clear presentation of results we first commented those related with the comparison of the main factors effects, and thereafter explained the particularities of the interaction in those cases this resulted significant. Anyway, text has been reviewed for a better understanding.

Table 5: The incubation medium for in vitro fermentation studies is well buffered to maintain a fixed pH (normally at 6.7-6.9, in our case around 6.3 to maintain a pH close to the expected fermentative conditions, L111-112). Consequently, the reference to the medium pH during fermentation is useful only to ensure an adjustment of the process to the desired incubation conditions, and potential variations in measured pH cannot be attributed to treatments as it depends on the buffer concentration included when making the media.

L314: Thanks for your indication regarding carotenoids and EE, a reference in this regard has been included. The comment relating the responses to the coefficient of variation has been removed.

Reviewer 2 Report

Dear authors, it seems to me that this manuscript has medium relevance in the scientific world. However, many points affect the quality of the manuscript.

General comments:

Correct the language and writing style. “milkingcows”: I think you were trying to write “lactating cows” or “dairy cows”

Abstract

The Simple Summary is better written than the Abstract. It is necessary to explicitly write the Objective and add the Conclusion. I think that by correcting the language, the writing style will be improved.

Introduction

The introduction is clear; there are precedents on these issues (Literature cited and their knowledge of the studies read). So what is the novelty/importance of your study?

What is your hypothesis?

On the other hand, the objective is not clear. The objective was evaluated in vitro rumen microbial fermentation and chemical composition? (Introduction) Evaluate opening time and processed method? (Title) Or genotype?(Statistic)

Material and methods:

This topic is a concern for me. 

First, the methodology is not fully described in terms of important data, such as the number of samples, experimental units, methods and processes, etc. 

Second, the number of samples tested, to me, has no power to find statistically significant results, bias results are more likely to be found. 

Third, the statistical evaluation: if the maize crop is the experimental unit, then your objective was to evaluate the genotypes of the crops?

Results:

This writing style needs to be improved. It is very repetitive.

I can't see the results as repeated measurements. For me, the results are only shown as comparisons of means. Where are the time charts?

Discussion:

Your discussion to me is a better description of the results and a comparative review of his found data and data from the literature. However, the topic of discussion is to explain chemically and biologically what happened to get those results and on this topic I can only see a fewspeculations, reviews and comparative studies. Improve it.

Conclusion

This conclusion needs to be improved. It is confusing, I recommend being objective/concise because in this way, it is a description of the results and/or comments of the authors.

Specific comments:

Line 24-26: Intricate text. Rewrite it.

Lines 69-70: forage stability or silage stability?

Lines 69-72: Rewrite it. In this form, it is difficult to read.

Lines 81-83: Describe the silage method used, the amount of inoculant, the type of silo, the number of silos, etc.

Line 88: How was the silage pelleted? Describe the method.

Line 153: Add the model used.

Line 165: Add the p-value.

Lines 168-169: Effects on ..?

Line 170: This type of text is repetitive with the table title and is not necessary. It is better to write the table number in parentheses at the end of the paragraph that describes the results of the respective table. Remove it.

Lines 170-173: This style of writing does not promote clear reading. Separate results by treatment and only report all together when there is interaction.

Lines 173-174: This is probably an explanation that can be used in the discussion topic. Also, for me, this is the last resort that should be used to explain a result.

Lines 179-180: See the previous comment for this type of text. Remove it.

Lines 180-189: Intricate text. Rewrite it.

Table 2 and other tables. I did not understand the letters that were used to show significant differences. The interaction is presented in another way and not in this way. I recommend looking for someone with a background in statistics to better present the results of engagement.

Lines 212-213: See the previous comment for this type of text. Remove it.

Lines 229-231Is this the description of the result of the interaction?

Lines 250-258: Ok, you compare your results with other results; what was the purpose? What is discussed here? Why were your results different on some points?

Lines 267-268: These lines are contradictory with the previous lines, either you explain it better or you avoid contradictions. Rewrite it.

Lines 280-282: If this was not expected in the current study, why did you use these lines to explain it?

Author Response

Thanks very much for your comments and suggestions.

Text has been reviewed by an English-speaking colleague in order to improve the English style.

Abstract: Text in the Abstract has been reviewed according to the comments from the Reviewer.

Introduction: Regarding the novelty of the work done, the major contribution is exploring the potential interest of processing (dehydration or pelleting after dehydration) of maize silage for improving the handling strategy of this feed, as this facilitates commercialisation. This has been now highlighted in text.

As stated at the end of the Introduction section, the objective was to study the effect of the ensiling length and the processing (dehydration or pelleting) of maize silage on chemical composition and rumen fermentation, as it is also stated in title. Genotype is out of the study, since the three tested varieties were considered only as experimental units (and consequently are not mentioned in the objectives).

Material and methods: We have tried to clarify the number of samples tested. Since 3 ensiling lengths, processed in 3 forms were assayed, only 9 treatments were considered; however, they were tested on 3 different maize crops, which give a total of 27 substrates. In order to avoid confusion, the word “treatments” in L90 has been changed for “substrates”.

Regarding representativity, it is worth considered that up to five samples from each of the 27 substrates (3 ensiling lenghts x 3 processing forms x 3 maize crops), were chemically analysed, and the average of these five was considered for statistical analyses; then for statistical comparison, the maize crops were considered as the experimental unit, and then treatment means were contrasted by ANOVA with an error term with 18 degrees of freedom. For fermentation studies, the average samples were assayed in three different runs (considered as blocks), so the number of repetitions was considerably high.

Testing maize genotypes was not an objective of the study, and this aim is not mentioned anywhere throughout the paper. Considering that the three varieties used were quite similar in terms of composition and degradability, maize varieties were considered as repetitions to improve the strength of the in vitro design for the study of the two factors of interest: ensiling length and processing form.

Results: We have tried to improve the writing of this section for an easier understanding; however, it has to be considered that all observed differences must be explained, and this is exhaustive regarding interactions.

Treatments are not compared statistically as repeated measurements, so I do not understand the Reviewer´s comment in this sense. We choose to present results of in vitro gas production in numeric form (Table 3) because a figure with 9 treatments should result difficult to highlight differences, and in this case should made explanation confusing.

Discussion: I am not sure to understand what the Reviewer means with this comment. There is a lack of information about dehydration and pelleting of maize silages (which is one of the novelties of the work), and most information on this topic is related with alfalfa, which is a totally different forage. This is what makes the Discussion quite speculative. In any case, we have tried to improve this section.

Conclusion: The conclusion section has been checked and corrected for a clearer writing. We consider that the mentions to potential implications in maize silage utilisation depending on the studied factors makes this section far from a mere description of results.

L24-26: Corrected

L69-72: Rewritten

L81-83: Information about silage conditions has been completed

L153: The statistical model has been included.

L165: In the former version, we did not include P-value when differences were not significant, since the P-rank for accepting differences was stated in subsection 2.4. In this case, it is included upon Reviewer´s request.

L168-169: This has been clarified.

L170, L179-180, L212-213: The suggestion from Reviewer 2 has been followed.

L173-174: This comment has been moved to Discussion

L180-189: These sentences have been checked.

L229-231: Yes, this is related to the interaction, as it is stated in the introductory sentence.

L250-258: The purpose of this paragraph is to characterise our feeds and to locate our results in the general scope reported in literature, as starting point for discussing the effects of the studied factors. Despite it is not a matter to exactly match with others, showing results that are within the expected range validates potential responses observed.

L267-268: The sentence agrees with the former comment. It is now rewritten to clearly show the statement.

L280-282: This is a part of the argument to support responses in starch concentration of maize silages. As this is not directly related to the type of maize varieties used, this sentence has been removed.

Reviewer 3 Report

The manuscript aimed to evaluate in vitro digestibility, methane concentration, and ammonia and volatile fatty acids concentration of maize silage after three ensiling lengths (3, 6 and 9 months) and under three processing forms (fresh, dried and pelleted). After opening the silo, the authors propose to dehydrate or to dehydrate and pellet the silage to facilitating the transport and distribution to a third part consumer (farm). The idea seems original, although in this paper the authors only evaluated the in vitro nutritive values, nor the feasibility of these processing in commercial field.

Simple summary:

L14: The author did not evaluate total tract digestibility, which refers to the use of animals. Indeed, the authors evaluated the in vitro digestibility.

Material and methods

The description of the treatments, methods and statistical design is very confused in several parts of the manuscript.

L80-85: According to the description, each silo had approximately 350 tons, is that alright? After opening, was the silo split into three parts of 116 tons each? Then 116 ton, remain as is (fresh), and ~232 tons were dried, after that 116 tons was ground to 4 mm and pelleted at 8 mm pellet size?

L89-90. According to "Simple summary" and "Abstract", treatments were defined as ensiling length (3, 6 or 9 months) and processing forms (fresh, dried and pelleted), which summed 9 treatments. Herein, the authors described 27 treatments, considering the three maize cultivars used. In line 156, the authors stated that maize crop (that I supposed was referred to the three maize cultivars) was considered as the experimental unit. What was the statistical design used and treatments?

L90-91: Was your treatment "processing conditions" or "processing forms" or "presentation forms"? Be consistent with terminology throughout the text.

L123-125: As described here, it seems that the authors used the concentration of neutral detergent fiber as a measure for in vitro true digestibility. Please, describe the formula used, not only the cited reference.

L133. Describe the acronym "CP" and "EE" at first use.

L142-143: At which moment, was the pH measured? After opening the silos and after dried and pelleted them?

L146: Change CH4 to CH4.

L154-156: What was the statistical design? What does the "n" mean: number of treatments or number of replicates? Was your treatment "processing conditions" or "processing forms" or "presentation forms"? Be consistent with terminology throughout the text.

L156-158: This analysis is valid just for in vitro gas production and DMd (Lines 115-117; Lines 122-123), but not for IVTD (Lines124-126), methane, ammonia and VFA (Lines 126-130). Describe the statistical models used.

Results

L165-167: At which moment was the pH measured? After opening the silos and after dried and pelleted them?

In general, tables were hard to follow and did not stand alone as is. Please find bellow a suggestion regarding Table 3 (attached file):

Discussion

L257-258: Please, verify if the mentioned in vivo OM total tract digestibility was 602 g/kg of dry matter. I assumed also that the unity here was also in dry matter basis.

L275-277: In terms of ruminant nutrition, increasing rumen starch degradation would increase efficiency of energy utilization. Please, explain more clearly and add reference why increasing rumen starch degradation will decrease efficiency of energy utilization.

L318-320. The increase in ADIN is a concern regarding processing methods with heating treatment, mostly because of Maillard reaction. I encourage the authors to discuss this matter, which might have influenced the lower in vitro gas production and digestibility of processed silages.

L330-332. If so, what are the fermented substrate and where they come from?

L340-342: Explain the relationship between lower volume of methane produced in 3m silages and the index of a lower contaminant potential.

References

Correct references 25 and 26; there are random numbers ("412" and "414") not related to the reference.

Author Response

We acknowledge the comment from Reviewer 3 regarding the possibility of focusing the paper to a commercial scope, but it is not possible to deal with all potential aspects in a single study. In our case, we considered that focusing the paper to nutrient characterisation provides consistency to the study itself.

L14: Amended

L80-85: The description is correct. Since the amounts are approximate, we consider is not necessary to emphasise more this aspect

L89-90: Since 3 ensiling lengths, processed in 3 forms were assayed, only 9 treatments were considered; however, they were tested on 3 different maize crops, which give a total of 27 substrates. Testing maize genotypes was not an objective of the study, and therefore maize varieties were considered as repetitions to improve the strength of the in vitro design for the study of the two factors of interest: ensiling length and processing form. In order to avoid confusion, the word “treatments” in L90 has been changed for “substrates”. 

L90-91: Throughout text, the terms “processing” or “processing forms” have now been used for defining treatments.

L123-125: The way to estimate IVTD is defined in text.

L133: Abbreviations for CP and EE are now explained in text.

L142-143: The silage pH was measured at the laboratory when samples were thawed for analyses. It is now specified in text (now in L173).

L146: Done

L154-156: The “n” indicates number of treatments. To avoid confusion, n has been changed for the subindex (i, j, k) used for each factor in the statistical model. As mentioned regarding the Reviewer´s comment on L90-91, the term “processing form” has been unified.

L156-158: To avoid confusion, this has been now clarified by changing the first part of the sentence.

L165-67: This has been answered regarding L142-143.

We agree with this comment regarding tables. Tables are complicated as they must be in agreement with the 3 x 3 experimental design. They show the length x processing means since this interaction resulted significant for many parameters (and differences among these means is what the Tukey test shows); however, for giving a complete picture, the P-values for main factors and the interaction are also shown in tables. We acknowledge the reviewer´s suggestion for the tables, but we think using two (upper and lower case) superscripts improves confusion, and in both cases they indicate the same because they have the same n for means comparison. In any case, tables presentation has been modified for their improvement.

L257-258: The value of 602 g/kg for total tract OMD is given by Khan et al. (2015) in their Table 2 for what they call “normal maize silages”. Our result is given in DM basis, as it is now specified in text regarding IVTD calculation but considering the low proportion of ashes in maize silages (Table 2) no major differences should be expected for the fact of being in vitro DM digestibility or OMD, being therefore comparable.

L275-277: What the sentence aims to indicate is that the energy from the starch reaching the small intestine is used more efficiently for the host than when it is used by rumen bacteria; therefore, if more starch is available for rumen fermentation a lower proportion should arrive to the duodenum. This is now further explained in text.

L318-320: A comment regarding the increase in ADIN has been included in Discussion.

L330-332: The sentence has been removed to avoid confusion.

L340-342: The whole sentence has been changed to avoid confusion.

References have been amended.

Round 2

Reviewer 2 Report

Dear authors, it seems to me that this manuscript has medium relevance in the scientific world. However, some points affect the quality of the manuscript.

General comments:

I believe that the authors wanted to evaluate many variables in order to present a quality work. As a result, they obtained data that is relevant. However, I think that the work could be much better used. Many results are not correctly described and therefore not correctly discussed. The interactions are not correctly described and therefore correctly discussed. 

Discussion:

What I wanted to say is that the topic of discussion is to explain chemically and biologically what happened to obtain these results.

To discuss the results is to explain why or how they were obtained (what influenced to obtain these results?). The literature review and comparative data (E.g. AAA obtained data similar to ours) are a complement to the discussion.

Specific comments:

Line 22-23: “maize cultivar” or “maize genotypes” instead of “maize crops”.

Line 24: “In vitro true digestibility instead of “true digestibility”

Line 32: Who is “it”?

Line 165: Need correction

Tables 2-5: I see that the interaction effect was described in the tables using letters. This is an incorrect way of displaying the results of the interaction. You need to add tables or figures with the decomposition of the interactions.

Author Response

Thanks very much for your comments. The fact that the 3 x 3 interaction resulted significant for many parameters is what considerably complicates the presentation of results. We consider that when that occurs (i.e., an interaction results significant) all effects must be explained, and this makes writing tedious and sometimes confusing. We have tried to make this section as clear as possible

We agree with the opinion form Reviewer 2 about how the Discussion must be structured. In our case, mentions to values from other authors have been included in the first paragraph of Discussion for support that our general results felt within the normal range expected for maize silages (for example, in terms of moisture, pH and chemical composition) to justify that results are trustable and to place our substrates within certain conditions. Thereafter, the potential factors causing the responses to the studied factors are discussed according to potential chemical or biological criteria, using comparisons of our results with others from literature in order to support the extracted hypotheses (for example, L294-302 in the present version). Among others, the use of arguments such as the increase of starch availability by solubilisation of the protein matrix with an increased length of ensiling (L273-293), the potential effect of silage pH after processing on an improvement of intake (L312-315), the effect of the dehydration/pelleting processing on fibre and a potential increase in ADIN (L315-326) and the implications of methane production from fermentation (L335-343) are also examples of such type of writing.

The specific comments have all been followed. Regarding the comment on L165, the writing of the reference no. 24 has been corrected.

Regarding the way of presenting interaction in tables: because of the existence of significant effects for the interaction for many parameters, the tables included in the manuscript show the 3 x 3 interaction means, and letters show such differences. The possibility of decomposing the interaction in figures (for example, a figure for the effect of processing within a specific ensiling length for every parameter that showed a significant interaction, which imply 13 figures) was initially discarded because it should extremely enlarge the paper. Instead, the presentation into partial tables was considered unnecessary since the tables actually included are in fact divided into processing forms (second column of each table). Even though letters in tables resulted a bit confusing in our case because of all the possible combinations, they indicate the result of the ANOVA (use of different letters for each factor should move to think they came from different statistical analyses, which is not true), and it is the most common way to present treatment differences.

Reviewer 3 Report

In general, the authors addressed the suggestions accordingly.

L278-281. It seems to have a syntax error in the sentence. Suggestion: "A release of starch in the rumen by a protein solubilisation during ensiling should increase rumen starch degradation and, thus, reduce the amount reaching the duodenum. Consequently, the change of starch digestion site should decrease the efficiency of energy utilisation."

Although, I still disagree with this comment.

Previous researchers have suggested that the ruminant small intestine has a limited capacity for starch digestion (Orskov, 1986; Owens et al., 1986; Swanson et al., 2002; Swanson et al., 2019). According to Harmon and Swanson (2020,

DOI:  https://doi.org/10.1017/S1751731119003136), "ruminants are limited users of small intestinal starch and that the low digestibilities in the small intestinal are likely the outcome of multiple factors that are only overcome by supplying small amounts of highly digestible substrate",

Therefore, the increase of rumen starch degradation in detriment of post-ruminal would increase the efficiency of energy utilisation.

Author Response

Thanks very much for your comments.

L278-281: The sentence has been amended accordingly. I agree with the comment from the Reviewer regarding the limited ability of ruminants for starch digestion at the small intestine. However, the classical comments of Orskov and Owens refer to high concentrate feeding conditions, where the input of starch to the duodenum is extremely high, as it occurs in high concentrate (with a high cereal proportion) feeding systems. We do not want to link our comment to any specific feeding situation, and simply refer that, for the ruminant, the efficiency of energy utilisation in the small intestine is higher than in the rumen because energy is directly used for the animal, instead of having to be fermented into VFA and then retransformed into pyruvate for being used in the energy cycle. The situation of a starch overload to the intestine might occur, for example, in intensive feeding beef systems such as those applied in Spain (up to 90% concentrate), but it is less probable in feeding milking cows (often within a range of 40 to 60% concentrate), where maize silage is commonly used.

Round 3

Reviewer 2 Report

Dear authors, it seems to me that this manuscript has relevance in the scientific world. I think the authors made the suggested changes; therefore, I recommend approval.

Author Response

Dear Editor,

thanks very much for considering our manuscript eligible for publication in Animals. We have followed tour recommendations and modified the final version in the following sense:

- A mention to the lack of differences among maize cultivars and an explanation of how this variation has been considered (as experimental unit instead of as a block) have been included in subsections 2.1 (now in L86) and 2.4 (now in L166-169) of Material and methods.

- The value of SEM has been further explained in the corresponding footnote of tables 2 to 5.